# Supramolecular Nanostructures Based on Perylene Diimide Bioconjugates: From Self-Assembly to Applications

**DOI:** 10.3390/nano12071223

**Published:** 2022-04-05

**Authors:** Nadjib Kihal, Ali Nazemi, Steve Bourgault

**Affiliations:** 1Department of Chemistry, Université du Québec, Montreal, QC H2X 2J6, Canada; kihal.nadjib@courrier.uqam.ca; 2Quebec Network for Research on Protein Function, Engineering and Applications, PROTEO, Quebec City, QC G1V 0A6, Canada; 3Centre Québécois sur les Matériaux Fonctionnels/Québec Centre for Advanced Materials, CQMF/QCAM, Montreal, QC H3A 2A7, Canada

**Keywords:** perylene diimide, self-assembly, nanostructures, peptides, oligonucleotides

## Abstract

Self-assembling π-conjugated systems constitute efficient building blocks for the construction of supramolecular structures with tailored functional properties. In this context, perylene diimide (PDI) has attracted attention owing to its chemical robustness, thermal and photo-stability, and outstanding optical and electronic properties. Recently, the conjugation of PDI derivatives to biological molecules, including oligonucleotides and peptides, has opened new avenues for the design of nanoassemblies with unique structures and functionalities. In the present review, we offer a comprehensive summary of supramolecular bio-assemblies based on PDI. After briefly presenting the physicochemical, structural, and optical properties of PDI derivatives, we discuss the synthesis, self-assembly, and applications of PDI bioconjugates.

## 1. Introduction

Mother Nature has always been a unique source of inspiration for the design of nanostructures and materials for application in various fields spanning from biomedicine to electronics. Living organisms are hierarchically built on the organized self-assembly of biomacromolecules, such as lipids, proteins, deoxyribonucleic acids (DNA), ribonucleic acids (RNA), and polysaccharides, controlled by a delicate balance of non-covalent interactions [1,2]. For example, the cellular plasma membrane results from the spontaneous organization of a diversity of lipids into a fluid and complex bilayer, while the exceptional mechanical properties of spider silk arise from the coordinated self-recognition of proteins. Over the last decades, biological macromolecules have been continuously harnessed as building blocks for the construction of (nano)structures with atomic-scale precision. Owing to their biocompatibility, functionality, and ability to undergo self-assembly, the conjugation of biological macromolecules to organic self-assembling molecules with interesting (photo)chemical properties has supported the construction of hybrid supramolecular architectures [3,4]. Of such small organic molecules, the family of perylene diimide (PDI; perylene-3,4:9,10-tetracarboxylic acid diimide) dyes has attracted tremendous attention. Particularly, PDI-based chromophores are known for their electron mobility, high fluorescence quantum yields, photo- and thermal stability, and semi-conductivity [5,6]. These properties benefit them in applications such as pigments, dye lasers, sensors, bioprobes, and photovoltaics [5,7,8]. Interestingly, the physicochemical, optical, and structural properties of PDI derivatives can be tuned by the modification of their substituents. Particularly, conjugating PDIs to biological macromolecules can allow the conception of supramolecular structures with unique architectures and properties. Albeit these chimeric PDI-biomolecules have been exploited for two decades, their self-assembly as well as the resulting morphologies remain difficult to predict from the monomeric building blocks, still precluding their full potential. This review aims at providing a comprehensive overview of PDI-bioconjugates by highlighting the most recent studies and by emphasizing the relationships existing between the physicochemical properties of the bioconjugates, the conditions of self-assembly, and the resulting supramolecular structures. While excellent reviews regarding PDI-based assemblies have been published [9,10,11], including the comprehensive reviews of Guo [9], and of Würthner [5], the present review focusing exclusively on PDIs conjugated to peptides and oligonucleotides will hopefully offer a novel perspective on this flourishing field of research.

## 2. Synthesis and Properties of PDI Derivatives

Owing to their stability and unique optical properties, organic π-conjugated systems constitute an important class of molecules for different applications in various fields, from biomedicine to (nano)material sciences. In particular, PDI and its derivatives are π-conjugated systems that have been widely exploited as building blocks for the preparation of stable and functional nanostructures with tailored optical and physicochemical properties [5,10,11]. To this end, several groups have developed innovative and reliable methods for the efficient synthesis of PDI derivatives and their subsequent self-assembly into tailored structures. In this first section, we will offer a brief overview of the main synthetic approaches commonly used to prepare PDI and its derivatives and we will present the optical, physicochemical, and self-assembly properties of PDI-based π-conjugated systems.

### 2.1. Physicochemical, Optical, and Self-Assembling Properties of PDIs

PDI derivatives have been recognized as an ideal π-conjugated system for chemical, colorimetric and fluorescent sensors [12,13], organic semiconductors and optoelectronic devices [14,15,16], phototheranostics, [17,18] as well as bioimaging and gene/drug delivering agents [19]. Structurally, the PDI molecule is composed of one rigid, planar, and stable perylene core with two *imide* groups at both ends of the polycyclic aromatic scaffold. The perylene core has 12 characteristic positions, known as the *bay* (1, 6, 7 and 12 positions), *ortho* (2, 5, 8 and 11 positions), and *peri* (3, 4, 9 and 10 positions). The attractiveness of PDIs as chromophores is associated with their chemical robustness, excellent thermal and photo-stabilities as well as their unique optical and electronic properties [9,10]. Soluble monomeric PDIs without substituents on the *ortho* and *bay* positions of the perylene core show prototypical UV-Vis absorption spectra characterized with three vibronic peaks and a typical mirror image emission spectrum with high fluorescence quantum yields in most organic solvents [5,20]. Interestingly, substitutions at the two *imide* positions have a negligible effect on the optical properties of PDIs, while they significantly modulate the self-assembling processes as well as the architecture of the resulting supramolecular structures [11,21]. In contrast, substituents at the *bay* positions have a considerable effect on the morphology as well as the optical, chiroptical [22], and electronic properties of PDI-based assemblies, such as exciton diffusion, charge transfer, and high fluorescence quantum yield with red-shifted emission bands, which are associated with applications in organic solar cells [23], organic semiconductor devices [24], organic light-emitting diodes [25], vapor sensors [26,27,28], and theranostic agents [29,30]. These *bay*-substitutions can lead to a distortion of the planar PDI that affects solubility and weakens the π–π staking interactions [11]. Consequently, the twisting of the PDI cores can be precisely tuned through *bay*-substitutions, leading to some control over the self-assembly and the desired morphology of the nano- and mesoscopic structures [30] and liquid-crystalline materials [31,32].

Generally, PDIs form weakly emissive H-aggregates in aqueous solutions via non-covalent π–π stacking interactions, leading to the linear orientation of aromatic systems, i.e., one top of each other (Figure 1). In contrast, the modulation of the planar structure by introducing bulky substituents into the *bay* positions of the perylene core can generate strongly emissive core-twisted self-assembled supramolecular structures [5], in which neighboring chromophores are oriented in a head-to-tail fashion, known as J-aggregates (Figure 1) [33,34].

### 2.2. Synthetic Strategies to Access Symmetrical PDIs

Perylene-3,4,9,10-tetracarboxylic dianhydride (PTCDA) is considered the parent compound of PDIs and is easily accessible through several well-defined multistep processes, including the one represented in Figure 1 [20,21,35]. In this example, the first step includes the synthesis of naphthalene-1,8-dicarboxanhydride via the oxidation of acenaphthene, followed by ammonia treatment to convert the anhydride to an *imide* functionality. *Imide* dimerization facilitated by molten potassium hydroxide provides perylene-3,4,9,10-tetracarboxylic diimide (PTCDI) that can be hydrolyzed using concentrated sulfuric acid at high temperature to obtain PTCDA. Imidization of PTCDA with primary amines or anilines, in imidazole and in the presence of anhydrous zinc acetate as catalyst, leads to the formation of appropriate symmetrically *N,N’-*substituted PDIs in high yields [20,21,35,36].

### 2.3. Synthetic Strategies to Access Asymmetrical PDIs

Asymmetrical PDIs can be obtained by the partial hydrolysis of PDIs to perylene monoimide monoanhydride followed by condensation of this mixed *imide*–anhydride compound with primary amines to access the desired asymmetrical PDIs, as illustrated in Figure 2 (Approach A) [21,37]. Moreover, the partial hydrolysis of PTCDA, which provides a mixed anhydride dicarboxylate salt, is another common approach to access asymmetrical PDIs via successive imidization reactions (Figure 2, Approach B) [21,35,37,38].

## 3. Peptide-PDI Bioconjugates

Over the last two decades, amino acids and peptides have been harnessed to control the solubility, optical properties and/or self-assembly of PDI derivatives by taking advantage of highly directional hydrogen-bonding interactions as well as other non-covalent interactions such as ionic and π−π interactions [39]. In addition, polypeptides are known for their biocompatibility, biodegradability and molecular specificity [40], ultimately supporting the use of peptide-based PDI assemblies for different biomedical applications. Chemically, peptides can be considered as linear polymers/oligomers assembled from the condensation of amino acid building blocks [41]. The 20 natural amino acids, as well as countless unnatural amino acids, allow virtually infinite combinations of sequences and offer an unlimited diversity of physicochemical and structural properties of the resulting polypeptide chains [42]. Herein, we present relevant examples of the use of amino acids and short peptide sequences to modulate the self-assembly of PDI derivatives into tailored nanostructures. The self-assembly of PDI–peptide conjugates and the resulting supramolecular morphologies are modulated by a fine balance of complex intermolecular interactions, such as PDI’s π–π stacking interactions and numerous non-covalent interactions involving side chains and the polyamide backbone, as well as by the conditions of the microenvironment, including solvent polarity, solution ionic strength, pH, and temperature [9,11,39].

### 3.1. Synthesis of Peptide-Conjugated PDIs

Amino acid– and peptide–PDI conjugates are commonly prepared by respectively introducing amino acids and short peptide sequences at one or both *imide* positions of PDI. Peptides are usually synthesized on solid support, which involves the attachment of the *C-*terminal amino acid to the polymeric resin by a covalent bond followed by the sequential addition of individual preactivated amino acids, and the final cleavage of the polypeptide from the solid support [43,44]. In contrast to solution synthesis, solid-phase peptide synthesis (SPPS) simplifies dramatically the purification steps between each reaction, thus reducing synthesis time and increasing yield. Consequently, not only does SPPS offer an efficient route towards automation, but also allows the routine access to high-molecular weight polypeptides of up to 50-residues [45]. Due to the insolubility of PTCDA as PDI precursor in common solvents used in SPPS, Kim et al. developed an alternative method to overcome this problem by initially conjugating the PTCDA to amino acids, accessing soluble PDI–amino acid derivatives [46]. Afterwards, the soluble PDI–amino acid derivative can be conjugated to the target peptide sequence that was elongated on the polymeric support by the common SPPS method (Figure 3) [46].

### 3.2. Sequence-Dependent Self-Assembly

Specific variations within peptide sequences have been used to control the self-assembly process of PDI conjugates and to modulate the morphology of the resulting nanostructures through a delicate balance of non-covalent interactions (hydrogen bonding, hydrophobicity, ionic bonding) involving specific residue side chains. Short dipeptides GX (where X = D or Y) were used to enhance the solubility and to modulate the self-assembly properties of PDI in polar organic solvents and aqueous solution [47]. It was observed that, by varying the X residue of the PDI–[GX]_2_ *bola*-amphiphile conjugates from hydrophobic Tyr to hydrophilic Asp, the balance between hydrogen bonding and π–π stacking interactions were altered, ultimately affecting the morphology of the assemblies and their optical properties. For instance, in aqueous sodium bicarbonate buffer (pH 10.8), dimethyl sulfoxide (DMSO), dimethylformamide (DMF), tetrahydrofuran (THF), or acetone, the symmetric PDI–[GY]_2_ formed chiral nanofibers, whereas PDI–[GD]_2_ assembled into achiral spherical aggregates in buffer and DMSO. In addition, PDI–[GY]_2_ formed a gel in DMF, while organogels were observed for the PDI–[GD]_2_ derivative in this polar aprotic solvent. An exhaustive study of structure-assembly relationships revealed how peptide’s physicochemical properties and length, asymmetric substitution at the *imide* positions, and stereocenter inversion can affect the thermodynamics of the self-assembly of peptide–PDI hybrid molecules [39]. A set of peptide–PDI conjugates were synthetized, all encompassing three units: (i) a glycine residue at the *N*-terminal position used as a low-steric hindrance linker; (ii) a central variable region composed of three L or D amino acids to evaluate the impact of increasing the peptide hydrophobicity and the role of stereocenters; and (iii) a *C-*terminal charged region composed of one to three Glu residues to enhance the hydrosolubility and examine the effect of the charge density on the assembly process (Figure 2) [39]. Moreover, to induce a strong amphiphilic character, one of the peptide sequences was replaced by a hydrophobic hexyl chain. It was observed that peptide hydrophobicity and an asymmetrical hexyl substitution induce significant changes on the aggregation thermodynamics of the bioconjugates (Figure 2a,b). In contrast, varying the peptide length, the *C-*terminal charged region length or the stereocenter inversion induced a significantly lower impact on the aggregation thermodynamics, while having an effect on the peptide-driven self-assembly of PDI nanofibers [39]. Overall, these studies revealed that the physicochemical properties of the residue side chains, the configuration of stereocenters, and the symmetric/asymmetric conjugation can be exploited to dictate the non-covalent interactions that drive the self-assembly of peptide-conjugated PDIs and the final morphology of the assemblies.

Inspired by the β-continuous interface of the bovine peroxiredoxin-3 protein, a short heptapeptide (IKHLSVN) was conjugated to PDI in order to control self-assembly into organic semiconductor nanostructures [48]. The designed self-assembling peptide encompassed three different regions: (i) a glycine or ethylamino linker attached to an *imide* position of the PDI to reduce steric hindrance between the PDI core and the peptide sequence, (ii) a β-sheet-forming peptide, and (iii) a terminal unit composed of glutamic acid residues to assist solubility and to trigger the assembly of peptide–PDI conjugates by pH jump. Two groups of peptide–PDI derivatives were prepared. The first group results from the symmetrical substitution of PDI with peptide sequences, whereas for the second group the PDI core was replaced with perylene *imide* bis-ester to generate asymmetrical derivatives. Furthermore, for one derivative of each group, the peptide core was attached to the PDI via the amino terminus using a glycine linker, while for the other derivative the peptide core was attached via the carboxyl terminus using an ethylamino linker. The symmetrically substituted PDI showed spectral profiles characteristic of monomers in DMSO. However, UV-visible spectral profiles of bis-ester-functionalized PDI displayed some aggregation, with predominantly monomeric species. In aqueous media, these peptide–PDI conjugates self-assembled into H-aggregate suprastuctures. Particularly, all peptide–PDI derivatives self-assembled into extensive fibril networks in aqueous solution, except for the bis-ester-functionalized PDI for which the peptide was attached via the *C-*terminal position. This compound formed amorphous, plate-like accretions. Furthermore, reversing the peptide sequence, i.e., *N-* to *C-*, for the symmetric derivatives led to the formation of short fibrils and thread-like assemblies instead of ribbon-like structures. This observation highlights the importance of the attachment mode of peptide to the PDI core towards self-assembly and final morphology of the assemblies.

Similarly, the *N*-(tetra (*L*-alanine) glycine)-*N′*-(1-undecyldodecyl) functionalized perylene-3,4,9,10-tetracarboxyl diimide was designed as an asymmetric amphiphilic derivative in order to elucidate how molecular-scale interactions govern the overall self-assembly process [49]. The oligopeptide block on one of the *imide* nodes of the PDI core provided aggregation directionality through hydrogen bonding and π−π stacking interactions. In chloroform, which was chosen to strengthen inter-peptide hydrogen bonds, this asymmetric amphiphilic PDI derivative adopted a right-handed helical arrangement due to the delicate balance between π–π stacking involving PDI cores and the network of hydrogen bonds between β-sheet-forming peptides. Interestingly, the addition of trifluoroacetic acid (TFA) to the self-assembling media, which was used as a hydrogen-bonding breaking agent, induced the transition of the nanofibers into small aggregates. These aggregates could be brought back into nanofibers by the addition of triethylamine (TEA), which was used to neutralize TFA, favoring the formation of H-bonding between the peptide blocks [49].

Computational simulation and experimental studies have been combined to understand the relationships between the sequence and the self-assembly process of π-conjugated peptides to ultimately predict the resulting supramolecular organisations and photophysical properties from the peptide sequence. For instance, it was shown that increasing the hydrophobicity of the closest residue attached to the PDI core can modulate the photophysical responses in aqueous solution via the conversion of J aggregates, or liquid-crystalline-type materials, to H-type aggregates [50]. In addition, the relationships between the resulting morphologies and the molecular structure of a small library of peptide–PDI derivatives bearing a variable number of *L*-alanine units as well as methylene, ethylene, and propylene spacers were investigated [51]. It was revealed that the number of *L*-alanine units in the β-strand peptide segments and the length of the spacer affected the morphology of the resulting suprastructures. In addition, it was shown, through molecular dynamic simulations, that there is a complex interplay between the translation of molecular chirality into supramolecular helicity and the inherent propensity for well-defined one-dimensional aggregation into β-sheet-like superstructures in the presence of a central chromophore [51]. Finally, a symmetric PDI–tripeptide conjugate, which was obtained by introducing a KPA tripeptide block at the 1 and 7 *bay* positions of PDI via a 2-(2-aminoethoxy)-ethoxyl linker, self-assembled into β-sheet nanohelices directed by hydrogen-bonding [52].These resulting supramolecular structures were particularly sensitive to thermal and ultrasound stimuli. For instance, upon heating/cooling and sonication of the peptide–PDI sample, an interconversion of the supramolecular chirality between left- and right-handed nanostructures was observed [52].

Except for glycine, which is achiral, α-amino acids have an S configuration and are designated as *L* using the Fischer configurational system [42]. It has been reported that the presence of chiral proximal residues in close proximity to the achiral PDI, for symmetrical peptide–PDI derivatives, influences π–π stacking interactions and induces helical chirality to the PDI core. However, when chiral residues were located distant to the PDI core, or when an isolated stereocenter was introduced in proximal distance of PDI, no effect of chirality was observed during self-assembly [53]. The designed peptide sequences with stereogenic positions and stereochemical configurations included three blocks: (i) an achiral glycine used as a spacer between PDI and the peptide, (ii) three variable residues forming the central block, and (iii) a terminal block of three ionizable glutamic acid residues to assist solubility and pH-triggered aggregation. It was observed that the self-assembly process is modulated by the β-sheet-forming potential of the peptide moieties and the π–π stacking interactions of PDI units. Interestingly, an inversion of the stereocenter within the proximal residues revealed chiral influence. In contrast, an asymmetrical peptide–PDI derivative obtained by the introduction of an alkyl chain at one of the amide nodes, generated an amphiphilic PDI conjugate and disrupted the chiral-mediated self-assembly [53].

Besides, it was observed that the symmetrical conjugation of FF dipeptide to the PDI core leads to a helical assembly due to the chirality of amino acids as well as the co-facial π–π stacking of PDI units [54]. Furthermore, it was shown that there is a close relationship between the translation of molecular chirality into supramolecular helicity and the one-dimensional assembly into well-defined β-sheet-like suprastructures [51]. Overall, these structure–assembly relationship studies have indicated that the morphology of the resulting peptide-conjugated PDI nanostructures can be, to some extent, controlled by modulating the peptide sequence. Moreover, the stereocenters embedded in the peptide backbone can be exploited, under specific conditions, to induce a chiral morphology to the resulting assemblies.

### 3.3. Solvent-Dependent Self-Assembly

As described above, several substituted water-soluble PDIs have been obtained by functionalizing the PDI core at their *imide* positions with amino acids and short peptides. Not only do the physicochemical properties of the conjugated moieties dictate the self-assembly behaviour of the PDI core as well as the morphology of the final aggregates, but the solvent also strongly influences the thermodynamics and kinetics of self-assembly [54,55,56]. It is generally assumed that the aggregation constant of PDI derivatives decreases with increasing solvent polarity [57]. The photophysical and aggregation propensities of PDI–[X]_2_ symmetrical derivatives, where X is a residue with an aromatic group (Y, W or F), were evaluated in various organic solvents [58]. This study revealed that all derivatives self-assemble into amorphous aggregates, excepted for PDI–[Y]_2_ that forms *J*-type aggregates in methanol. Although it is still unclear why this effect manifested only in methanol, the authors suggested that the formation of J-aggregates may be possible due to the network of hydrogen bonds involving the hydroxyl groups of tyrosine residues and the carbonyl groups of the PDI core. In pyridine and acetone, PDI–[F]_2_ and PDI–[Y]_2_ showed a higher propensity to aggregate than PDI–[W]_2_, however, the origin of this tendency was unclear. Furthermore, the NMR data indicated a large degree of aggregation in DMSO for all three PDI–peptide derivatives, although the absorption and the fluorescence spectra were both characteristics of soluble and monomeric PDIs [58]. In another study, it was reported that the relatively polar nature of chloroform, in contrast to THF and DMF, facilitates the formation of intermolecular H-bonding of PDI–[F]_2_, which ultimately leads to *J*-aggregates [59]. In chloroform, PDI-[F]_2_ assembled into vesicular suprastructures through the formation of right-handed helix, involving intermolecular H-bonding in lateral and π–π stacking in longitudinal growth directions. As shown in Figure 3, the increase in solvent polarity correlated well with a decrease in the diameter of the vesicles assembled from the PDI–[F]_2_.

Similarly, the self-assembly of the symmetrical PDI–[FF]_2_ conjugate and the morphology of the resulting suprastructures were dramatically modulated by the polarity of the solvent [55]. For instance, fibrillar nanostructures were obtained in relatively non-polar solvents, such as THF and CHCl_3_, while in more polar solvents (HFIP, MeOH, ACN, acetone), spherical morphologies were observed, for which the diameter correlates inversely with the polarity of the solvent (Figure 4) [55]. By density-functional theory (DFT) calculations combined with experimental studies, the kinetic and thermodynamic parameters driving the self-assembly of PDI–[FF]_2_ were investigated in the THF/water mixed solvent system [54]. In THF, PDI–[FF]_2_ formed right-handed helical nanofibrils under kinetic control, while in 10% THF the helicity switched to left-handed orientation governed by thermodynamics, leading to the formation of nanorings [54].

By harnessing *N,N’*-α-carbobenzyloxy-*L*-lysine-functionalized 1,6,7,12-tetrachloro-perylene diimide, it was further demonstrated that the polarity of the solvent strongly modulates the morphology of the nanostructures assembled from PDI–amino acid chimeric derivatives [56]. These PDI derivatives were assembled via a phase transfer strategy for which the compounds were first dissolved in acetone followed by the gradual addition of water to reach the desired ratios. When the -COOH group of the lysine was placed five carbon–carbon bonds away from the *imide* node position of PDI (i.e., PDI-A), increasing the H_2_O content up to 55% did not affect the morphology of the suprastructure, but caused an important decrease in nanobelt length with an increase in their width. (Figure 5a). Interestingly, conversion into spherical aggregates was observed when the water percentage increased up to 75% (Figure 5c). The effect of the solvent nature on PDI–peptide self-assembly was attributed to a delicate balance between reassembly at low H_2_O fractions to minimize the system’s free energy (thermodynamic control) and fast nucleation (kinetics control) in presence of high water content to adjust toward the optimal conformation. Moreover, by moving the carboxylate of the lysine directly on the next carbon attached to the PDI’s *imide* node (i.e., PDI-B), it was observed that the position of the COOH group is a critical factor governing the nanostructure morphology. The PDI-B derivative assembled into condensed, layered structures characterized by a mixture of nanosheets and spheres in mixed acetone/water solution. The presence of small spheres became progressively more prevalent as the water content increased (Figure 5d–f). It was suggested that the formation of these ill-ordered structures was triggered by the steric hindrance of the carboxylate next to the *imide* node, which obstructs intermolecular H-bond formation between neighboring -COOH groups and deteriorates their long-range ordering [56].

### 3.4. PH-Dependent Self-Assembly

In addition to the peptide sequence and solvent polarity, pH plays a critical role in dictating the hierarchical organization of amino acid–PDI and peptide–PDI derivatives in aqueous solution. Using A-, H-, F-, and V-functionalized PDIs, it was reported that symmetrical PDI-[A]_2_, PDI-[H]_2_ and PDI-[V]_2_ conjugates formed worm-like micelles at high pH (~10), whereas transparent gels were obtained at low pH [60]. The PDI-[F]_2_ conjugate formed a turbid gel at low pH and a disordered structure upon drying of the solutions in air. Films obtained by drying solutions and gels of the other residue–PDI derivatives showed similar structures, with thin entangled fibers that are less aligned in xerogel state [60]. Other studies have also explored the role of pH in controlling the self-assembly of the symmetrical PDI–[X]_2_ derivatives. While the carboxylate groups at both ends provide good solubility in basic aqueous solution, aggregation can be triggered by the protonation of the carboxylate groups at pH values below their pKa [61,62]. Moreover, addition of carboxylic acid groups at the two terminal positions was also used to increase the hydrosolubility of a bola-amphiphilic system composed of a PDI core conjugated on their *imide* positions with phenylalanine via an oligomethylene glycol spacer [63]. Designed to self-assemble through π–π staking, H-bonds and van der Waals interactions, this peptide–PDI conjugate self-assembled into hydrogels in phosphate buffer within a pH range of 7−9 and showed interesting photoswitching properties in xerogel state, characterized by a nanofibrous network [63]. A bola-amphiphilic system composed of a PDI core conjugated at its *imide* positions with histidine (PDI–[H]_2_) demonstrated a pH-responsive protonation-deprotonation that controlled non-covalent interactions between the imidazole ring of adjacent monomers and modulated their self-assembly behavior in water [64]. At pH 10, PDI–[H]_2_ self-assembled into long, thin fibers with a 12 ± 1 nm width. At pH 7, thick fiber bundles with a 22 ± 1 nm width were observed, which evolved into interconnected belt structures with a 120 ± 5 nm width at pH 2 [64]. In summary, according to the peptide sequence, the pH of the medium can be exploited to control the net charge of the system and the non-covalent interactions between side chains in order to drive self-assembly of the peptide-conjugated PDIs and the resulting nanostructures.

### 3.5. Concentration-Dependent Self-Assembly

Numerous studies have shown that the concentration of peptide-PDI conjugates can also influence their self-assembly as well as the final supramolecular architecture by modulating enthalpically and/or entropically driven self-recognition processes. Interestingly, it was shown that the symmetrical PDI-[F]_2_ derivative exhibited distinct concentration-dependent photophysical properties in different organic solvents, such as THF, DMF, and CHCl_3_ [59]. Increasing PDI–[F]_2_ concentration from 0.45 mM to 14.6 mM in CHCl_3_ caused a 43 nm red shift of the fluorescence emission peak, indicative of formation of J-aggregates. As expected, the self-assembly of PDI-[F]_2_ was stimulated with increasing its concentration. Additionally, the self-organization propensity of PDI-[F]_2_ derivative correlated closely with the polarity of the solvent (CHCl_3_ > THF > DMF). Another study on concentration-dependent aggregation property of symmetrical water soluble PDI-[X]_2_ systems (where X = T or D) has shown an inversion of the maximum absorption intensities and gradual photoluminescence quenching with increasing concentrations, from 10 μM to 250 μM [61]. The *N,N’*-α-carbobenzyloxy-*L*-lysine-functionalized 1,6,7,12-tetrachloroperylene diimide compound formed short nanofibrils at low concentrations ranging from 100 to 300 μM, whereas it self-assembled into 16 μm long sheet-like materials at concentrations over 800 μM. Further concentration increases up to 1 mM induced the formation of a mixture of nanotubes and large nanosheets, highlighting how the concentration of PDI-bioconjugates can dramatically modulate the resulting architectures [56].

## 4. Oligonucleotide–PDI Bioconjugates

Oligonucleotides are well known for their capacity to self-associate in a programed manner into organized supramolecular structures of various sizes and shapes, which offer exceptional opportunities in nanomedicine and nanotechnology [65,66,67]. DNA-based technology exploits non-canonical base pairing to easily access a large diversity of tailored three-dimensional DNA complexes, also known as DNA origami [66,67,68]. As previously described for peptides, oligonucleotide sequences have been exploited to modulate the solubility, optical properties, pathways of self-assembly, and/or supramolecular structures of PDI derivatives by means of specific base-pairing as well as hydrophobic and π–π interactions between the PDI core moieties and the oligonucleotide chains. In this section, we will briefly present the main synthesis routes to access oligonucleotide–PDI conjugates before introducing the key studies regarding their self-assembly into supramolecular structures.

### 4.1. Synthesis of Oligonucleotide–PDI Bioconjugates

Many synthetic approaches used to incorporate PDI as DNA base surrogate have been described to date. The modified oligonucleotides can be synthetized via automated phosphoramidite chemistry and be used as DNA building blocks that allow incorporation of the PDI moiety as an artificial nucleoside surrogate, either at the 5‘-terminus or at internal positions of duplex DNA [69]. This approach starts with the preparation of the mixed bisimide from the protected enantiomerically pure (S)-aminopropane-2,3-diol, followed by the coupling of the phosphiteamide group to the free hydroxy functionality of the mixed diimide intermediate, as illustrated in Figure 4.

A convenient click chemistry approach has also been used for the preparation of PDI–oligonucleotide conjugates. The reaction was carried out in a mixture of water and DMSO by means of a copper-catalyzed Huisgen [3 + 2] cycloaddition reaction involving alkyne and azide, as shown in Figure 5 [70].

### 4.2. Structure-Dependent Self-Assembly of DNA–PDI Bioconjugates

The oligonucleotide sequence, as well as its length and the positioning of the incorporated PDI(s), are known to significantly modulate the self-assembly process as well as the properties and the morphologies of oligonucleotide–PDI archi. For instance, it was observed that when the PDI chromophore is internally inserted as a synthetic nucleoside surrogate, it favors strong π-stacking interactions with the adjacent DNA base pair. In contrast, dimerization of two whole DNA duplexes is induced when the PDI is attached to the 5‘-end [69]. Interestingly, it was observed that the incorporation of multiple PDI moieties into the DNA backbone generates folded nanostructures that exhibit unique hyperthermophilic properties (Figure 6) [71]. It has been proposed that hydrophobic attractions between PDI moieties play an important role in stabilizing the folded structure at high temperatures, and that the DNA loops do not contribute to the inverse-temperature folding behavior [71].

Furthermore, it was reported that the parent PDI and pyrrolidine-substituted PDI pair are comparable to the natural base pairs and can lead to the formation of stable and stacked heterodimers within DNA duplexes [72]. These modified DNA duplexes were obtained by using enzymatically generated abasic sites that create reactive sites for PDI. The influence of the hydrophobic π–π stacking of PDI groups and base-pairing of oligo tails on the self-assembly behavior of oligonucleotide-conjugated PDI have been explored by varying the length and composition of the oligonucleaotides [73]. Helical as well as nonhelical fibers were obtained from π-stacked PDI groups conjugated with pendent short oligo tails in aqueous solution. The oligonucleotide tails provided solubility in water, whereas the hydrophobic π–π interactions involving PDIs governed self-recognition. In contrast, long oligonucleotide tails led to a diversity of large and ordered assemblies governed by means of base-pairing as well as hydrophobic π–π stacking induced by PDI head groups [73].

Molecular dynamics simulation was employed to understand the self-assembly dynamics and kinetics of the oligonucleotide backbone covalently linked to PDI moieties [74]. It was shown that the mechanism of formation of PDI trimers requires two steps. According to this model, two PDI molecules initially stack into a dimer and the third PDI moiety is then aligned on top of this dimer to finally form a trimer. This study also revealed that each PDI pair interrelates through attractive van der Waals interactions and repulsive electrostatic interactions to drive their self-assembly into ordered structures [74]. Fascinatingly, up to six PDI dyes were covalently incorporated into an oligonucleotide sequence, yielding a DNA duplex characterized by excimer-type fluorescence [75]. The presence of thymines in abasic opposite site of the counter strand impacted PDI-based hydrophobic interactions and the authors proposed a zipper-like recognition motif of the PDI hexamers formed inside the DNA duplex (Figure 7) [75].

Moreover, the 1,7-dibromo-PDI derivative was 5′-conjugated to a short coding sequence derived from the human telomeric and it was observed that the PDI moieties drive dimerization of these DNA strands [76]. The topology and stoichiometry of the structures assembled from this PDI–oligonucleotide were modulated by the ionic strength of the solution as well as by the concentration of the bioconjugates. Oligonucleotide strands quickly self-assembled into G-quadruplexes in desalted solutions, whereas Q-assemblies were obtained in the presence of Na^+^ or K^+^. In this system, the PDI moiety formed dimers instead of extended aggregates, allowing the formation of antiparallel Q species that ultimately converted into parallel species [76].

In another study, PDI was used as a hairpin linker to design short synthetic DNA hairpins in which the PDI core and a guanine–cytosine (G–C) base pair, which serves as a hole trap, are systematically separated by adenine–thymine (A–T) base pairs of various lengths (Figure 8) [77].In this elegant study, it was observed that the presence of stacked PDIs leads to the dimerization of the PDI–oligonucleotide hairpins in buffer solutions and their photoexcitation resulted in subpicosecond formation of a lower exciton state, followed by the formation of an excimer-like state.

### 4.3. pH-Dependent Self-Assembly of DNA–PDI Bioconjugates

pH modulation offers an interesting approach to control the self-assembly behavior of PDI molecules conjugated to oligonucleotides as well as the resulting morphology of the chimeric nanostructures. In order to elucidate the roles of PDI aggregation on the formation of G-quadruplex DNA, two symmetrical PDI derivatives respectively bis-substituted in *imide* position by piperidino–ethyl or morpholino–propyl groups were synthetized and their pH-dependent self-assembly was investigated [78]. Under acidic conditions, the PDI–DNA derivatives were not aggregating and were bound avidly to both duplex and G-quadruplex DNA. In sharp contrast, under basic conditions, the ligands extensively aggregated with a high degree of selectivity to bind the G-quadruplex DNA over the DNA duplex. The pH-dependent self-assembly of these PDI derivatives differs according to the nature of the amide substituents. For instance, the more basic piperidino side chain requires higher pH (> 8–9) compared to the derivative bearing the less basic morpholino group, which extensively aggregated at pH > 7–8 [78]. Thus, the pH-dependent self-assembly of DNA-PDI bioconjugates can be exploited to control non-covalent interactions and the resulting morphologies of the nanomaterials.

### 4.4. Temperature-Dependent Self-Assembly of DNA–PDIs

As for double-stranded DNA, temperature plays a critical role in the self-recognition of PDI–oligonucleotide conjugates. The temperature-dependent self-assembly of synthetic DNA dumbbells having 6 to 16 A–T base pairs connected at their extremities by two PDI linkers at the *imide* nodes was investigated [79]. This PDI-linked bis(oligonucleotide) conjugate was present predominantly as a monomer at room temperature in low-ionic-strength aqueous buffer. Upon heating, a stack of PDIs and a fusion of the base pairs was observed as noticed with the UV and fluorescence spectra (Figure 9a). Transition from DNA capped hairpin to collapsed dumbbell structures with intramolecular π–π stacking of PDIs involved an intermediate state for which all base pairs must be dissociated (Figure 9b). Molecular dynamic simulations highlighted electronic interactions between PDI moieties and the adjacent base pair in the DNA for the capped hairpin while π–π interactions between parallel PDIs were inferred for the dumbbell form [79].

The PDI scaffold was also used as a hairpin linker to produce a hairpin-forming bis(oligonucleotide) conjugate that remains predominantly in its monomeric form at room temperature in low-ionic-strength aqueous solution, whereas it dimerizes to head-to-head hairpins under high salt concentrations (>50 mM NaCl) [80]. This effect of salt on the assembly state was attributed to the increase in cation condensation in the hairpin dimer vs. monomer. In contrast, upon heating and in the presence of a low salt concentration, the hydrophobic association between the two PDI units was disrupted, leading to the formation of a monomeric hairpin followed by dissociation of base pairs to reach a random coil structure (Figure 10) [80].

## 5. Potential Applications of PDI Bioconjugates

### 5.1. Applications of Peptide–PDI Bioconjugates

As for other PDI-based derivatives, peptide-functionalized PDI conjugates were developed for a variety of applications spanning from microelectronics to biomedicine. A number of studies have reported interesting electronic and optoelectronic properties of PDI–peptide assemblies and showed that these hybrid nanomaterials can be used to fabricate organic light-emitting diode devices and bioelectronic materials [48,59,81]. Interestingly, a sensor probe for Pd^2+^ and CN^−^ was designed by incorporating two pyridine groups in the PDI core through an Asp residue linker [82]. The asymmetrical PDI-based probe showed high selectivity toward Pd^2+^ ions over other metal cations including Mg^2+^, Sr^2+^, Al^3+^, Cr^3+^, Pb^2+^, Mn^2+^, Fe^3+^, Co^2+^, Ni^2+^, Zn^2+^, Cd^2+^, Hg^2+^ and Cu^2+^, with the formation of Pd^2+^-ligand complexes triggering the aggregation process and quenching PDI’s emission. The presence of CN^−^ ions stimulated disaggregation of the Pd^2+^-induced assemblies (Figure 11). Particularly, this system was found to be extremely selective for CN^−^ over other anions, including OAC^−^, ClO_4_^−^, HSO_4_^−^, H_2_PO_4_^−^, SCN^−^, BF_4_^−^, PF_6_^−^, NO_3_^−^, OH^−^, F^−^, Cl^−^, Br^−^, I^−^, SO_4_^2−^, and PO_4_^3−^. This sensor probe that efficiently measures the concentration of Pd^2+^ and CN^−^ with detection limits at 0.55 ppb and 0.226 ppb, respectively, constitutes a relevant example of potential applications for PDI-peptide conjugates.

In addition, PDI–amino acid derivatives were used as fluorescent probes to selectively detect anions [6]. It was demonstrated that PDIs respectively functionalized with *L*-alanine, *L*-glutamic acid, *L*-phenylalanine or *L*-tyrosine exhibit selectivity and high sensitivity for the anions F^−^ and OH^−^, which involve a synergetic effect of H-bonding and anion–π interaction. It was revealed that films obtained by drying solutions and gels of the *L*-alanine, *L*-histidine, *L*-phenylalanine and *L*-valine functionalized PDIs are photoconductive, and this photoconductivity correlates with the formation of the radical anion [60]. Interestingly, the photoreduction of PDI gelator was associated with the formation of a stable radical anion that induces a change in the packing of the PDI assemblies. This ultimately led to structural modification of the fibrous network, and changes in the rheological properties of the gels [81].

Pyrophosphate (PPi) plays crucial roles in numerous biochemical reactions such as the inhibition or activation of some enzymes, cellular metabolism, protein and nucleic acid synthesis, cell proliferation and cellular iron transport, but abnormal levels can lead to physiopathological conditions [83]. A PDI–peptide derivative was developed as a selective and sensitive fluorescent probe for detecting PPi by exploiting the on/off fluorescence. Complexed to cupric acid ion, the PDI–[GD]_2_ self-assembled into PDI–[GD]_2_/Cu^2+^ aggregates, which led to the fluorescence quenching of the PDI units [84]. The displacement of Cu^2+^ by pyrophosphate, for which the PDI–[GD]_2_ had higher affinity, induced the disassembly of the aggregates and fluorescence recovery in which the intensity correlates directly to the concentration of pyrophosphate in solution [84]. Moreover, symmetrical cysteine-modified PDI (PDI–[C]_2_) was developed as a simple, low-cost and selective sensing probe for mercury ions, a toxic heavy metal pollutant, in aqueous media [85]. Besides, it was revealed that the thin film formed by the assembly of symmetrical PDI conjugated to histidine (PDI–[H]_2_) can be used as a reliable, cost-effective, and selective sensor device for the detection of NH_3_ vapors under ambient conditions [86]. A simple strategy was developed for the detection and rapid clearance of bacterial lipopolysaccharides (LPS) by combining a specific targeting ligand based on a PDI-conjugated LPS-recognition peptide, with magnetic Fe_3_O_4_@SiO_2_ core–shell structures [87].

Interestingly, PDI-peptide conjugates have been evaluated as supramolecular structures for photocatalytic hydrogen production. It was observed that the self-assembly of PDI–[H]_2_ is needed for hydrogen evolution, i.e., the production of hydrogen through water electrolysis, which occurs at pH 4.5 [62]. Furthermore, another study has demonstrated the stability, phototoxicity and biocompatibility of PDI–[ARGD]_2_ symmetrical derivatives, suggesting potential usage in the field of photodynamic therapy [46]. Finally, a biocompatible PDI[H]_2_ probe was described as a unique system towards the modulation of the amyloid fibril formation process [88]. This probe co-assembled with amyloid-β peptide (Aβ), which is associated with Alzheimer’s disease, via H-bonding that led to the enhancement of the π−π interactions between Aβ and PDI–[H]_2_. This interaction accelerated the aggregation process of Aβ into large micron-sized co-assembled structures. Since the oligomeric forms of amyloidogenic peptides are reported to have higher toxicity as compared to the fibrillar aggregates [89,90], these PDI conjugates could lead to potential design and development of drugs targeted toward Alzheimer’s disease treatment [88].

### 5.2. Applications of Oligonucleotide-Conjugated PDIs

Numerous studies have shown that oligonucleotide-conjugated PDIs can be used in biological applications and imaging tools owing to their self-assembly and programmability. For instance, high quenching efficiency of the parent PDI/pyrrolidine-substituted PDI heterodimers, which were arranged in a DNA stack, was exploited to design fluorescent probes toward specific nucleic acid sequences [71]. In this fluorescent reporter, the closed form of the molecular beacon (MB) induces π–π interactions of the pair in the stem region of the molecular beacon sequence, which leads to fluorescence quenching (Figure 12). This closed form is converted to the open form in the presence of a specific target oligonucleotide strand that is complementary to the loop region of the molecular beacon, leading to fluorescence recovery of parent PDI (PH)/pyrrolidine-substituted PDI (PN) heterodimers [72].

The binding selectivity of cationic PDI to thymine–thymine (T-T) mismatches and the formation of PDI dimer at the mismatch site were exploited to design biosensors [91]. In fact, it was observed that the presence of Hg^2+^ ions produced switching of cationic PDI emission from excimer to monomer due to the disruption of the dimeric form and the conversion of T-T mismatch into T–Hg–T base pair in DNA. Thus, the binding and optical properties of cationic PDIs can be exploited to develop fluorescent probes for T-containing DNA mismatches as well as for Hg^2+^ ions [91]. Moreover, a light-up fluorescent sensor for nucleic and ribonucleic acids with nanomolar sensitivity was developed by conjugating a PDI derivative, harboring an alkoxy group at the *bay* position, to an artificial pocket made by replacing nucleosides with deoxyribospacers inside the DNA structure [92]. In contrast to the assembled form, the monomeric form of PDI linked to the DNA pocket exhibited distinct photochemical properties.

Furthermore, it was proposed that PDI–triplex conjugates can be used to specifically target single-stranded nucleic acid sequences and to modulate genetic expression by the formation of complexes with segments involved in transcriptional processes [93]. In addition, self-assembled DNA monolayers conjugated with PDI base surrogate were used to develop a redox-based probe [94]. It was observed that the rate of electron transfer, mediated by stacks of DNA base pairs, is dependent on the stability of the DNA bridge and the distance between the PDI moiety and the electrode surface [94]. Finally, a cyclic PDI ligand was designed and evaluated as an anticancer drug owing to its high selectivity and efficiency in inhibiting telomerase activity [95].

## 6. Conclusions and Perspectives

PDIs are some of the most relevant organic dyes that have been largely studied and extensively used, owing to their chemical robustness, thermal and photo-stability as well as their physicochemical and optoelectronic properties. In addition, once rendered amphiphilic, PDI-based molecules undergo self-organization to form nanostructures of varying morphologies. However, the precise control of their molecular self-assembly into ordered supramolecular nanomaterials remains a highly active research area. Conjugation of PDIs to oligonucleotides and peptides has opened new avenues for the design of nanomaterials with unique structures, properties, and functionalities. Such an achievement is largely due to the advances made in the synthetic approaches to (a)symmetric PDI derivatives, and the development of cutting-edge technologies for characterization. The self-assembly process of these PDI-bioconjugates as well as the morphology of the resulting superstructures are dictated by internal non-covalent interactions and numerous external microenvironment conditions. This balance of internal and external factors complexifies the prediction of the morphology of the resulting supramolecular architectures. Nonetheless, as highlighted in this review, recent advances regarding the controlled assembly of peptide–PDI and DNA–PDI bioconjugates will most likely lead to a plethora of innovative functional biomaterials that can be exploited for various biomedical and nanotechnological applications.

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
