# Peer review of "Supramolecular Nanostructures Based on Perylene Diimide Bioconjugates: From Self-Assembly to Applications"

_nanomaterials, 2022, doi:10.3390/nano12071223_

Round 1

Reviewer 1 Report

This work offers a comprehensive summary of supramolecular bio-assemblies based on PDI. After briefly presenting the physicochemical, structural, and optical properties of PDI derivatives. The synthesis, self-assembly and applications of PDI bioconjugates are also discussed. The review is properly described and discussed. Thus, I recommend this manuscript for publication in Nanomaterials after some revisions towards the following points:

  1. It is very necessary to add some new references based on self-assembling properties of PDIs(J. Am. Chem. Soc. 2020, 142, 7092; Langmuir  2020, 36, 21, 5954; Journal of the American Chemical Society  2022, 144, 5, 2360; The Journal of Physical Chemistry B  2009, 113, 16, 5376).
  2. The authors should enrich the content of potential applications of PDI bioconjugates, such as solar cells (ACS Applied Energy Materials 2019, 2, 5, 3918; ACS Applied Electronic Materials 2019, 1, 8, 1590;ACS Applied Materials & Interfaces 2018, 10, 17, 14986), liquid crystalline (The Journal of Physical Chemistry Letters 2016, 7, 7, 1327), biomedical (ACS Appl. Bio Mater. 2020, 3, 1607), and so forth.
  3. In the conclusion and perspectives section, the author needs further condense and summarize.

Author Response

We thank the reviewer for his positive and constructive comments and suggestions.

It is very necessary to add some new references based on self-assembling properties of PDIs(J. Am. Chem. Soc. 2020, 142, 7092; Langmuir  2020, 36, 21, 5954; Journal of the American Chemical Society  2022, 144, 5, 2360; The Journal of Physical Chemistry B  2009, 113, 16, 5376). The authors should enrich the content of potential applications of PDI bioconjugates, such as solar cells (ACS Applied Energy Materials 2019, 2, 5, 3918; ACS Applied Electronic Materials 2019, 1, 8, 1590; ACS Applied Materials & Interfaces 2018, 10, 17, 14986), liquid crystalline (The Journal of Physical Chemistry Letters 2016, 7, 7, 1327), biomedical (ACS Appl. Bio Mater. 2020, 3, 1607), and so forth.

As requested by the reviewer, we included the following properties and applications:

Line 82: « chiroptical »24

(24) Liu, B.; Böckmann, M.; Jiang, W.; Doltsinis, N. L.; Wang, Z. Perylene Diimide-Embedded Double [8]Helicenes. J. Am. Chem. Soc. 2020, 142 (15), 7092–7099. https://doi.org/10.1021/jacs.0c00954.

Line 85: « organic semiconductors devices »26

(26) Zhang, L.; Song, I.; Ahn, J.; Han, M.; Linares, M.; Surin, M.; Zhang, H.-J.; Oh, J. H.; Lin, J. π-Extended Perylene Diimide Double-Heterohelicenes as Ambipolar Organic Semiconductors for Broadband Circularly Polarized Light Detection. Nat Commun 2021, 12 (1), 142. https://doi.org/10.1038/s41467-020-20390-y.

Lines 85-86: « organic light-emitting diodes »27

(27) Dayneko, S. V.; Rahmati, M.; Pahlevani, M.; Welch, G. C. Solution Processed Red Organic Light-Emitting-Diodes Using an N -Annulated Perylene Diimide Fluorophore. J. Mater. Chem. C 2020, 8 (7), 2314–2319. https://doi.org/10.1039/C9TC05584C.

Line 86: « vapor sensor »28

(28) Wang, J.; He, E.; Liu, X.; Yu, L.; Wang, H.; Zhang, R.; Zhang, H. High Performance Hydrazine Vapor Sensor Based on Redox Mechanism of Twisted Perylene Diimide Derivative with Lower Reduction Potential. Sensors and Actuators B: Chemical 2017, 239, 898–905. https://doi.org/10.1016/j.snb.2016.08.090.

Moreover, as recommended by the reviewer, we included the following additional references:

(8) Abd-Ellah, M.; Cann, J.; Dayneko, S. V.; Laventure, A.; Cieplechowicz, E.; Welch, G. C. Interfacial ZnO Modification Using a Carboxylic Acid Functionalized N-Annulated Perylene Diimide for Inverted Type Organic Photovoltaics. ACS Appl. Electron. Mater. 2019, 1 (8), 1590–1596. https://doi.org/10.1021/acsaelm.9b00328.

(16) Yuen, J. D.; Pozdin, V. A.; Young, A. T.; Turner, B. L.; Giles, I. D.; Naciri, J.; Trammell, S. A.; Charles, P. T.; Stenger, D. A.; Daniele, M. A. Perylene-Diimide-Based n-Type Semiconductors with Enhanced Air and Temperature Stable Photoconductor and Transistor Properties. Dyes and Pigments 2020, 174, 108014. https://doi.org/10.1016/j.dyepig.2019.108014.

(17) Song, C.; Liu, X.; Li, X.; Wang, Y.-C.; Wan, L.; Sun, X.; Zhang, W.; Fang, J. Perylene Diimide-Based Zwitterion as the Cathode Interlayer for High-Performance Nonfullerene Polymer Solar Cells. ACS Appl. Mater. Interfaces 2018, 10 (17), 14986–14992. https://doi.org/10.1021/acsami.8b01147.

(18) Yang, Z.; Chen, X. Semiconducting Perylene Diimide Nanostructure: Multifunctional Phototheranostic Nanoplatform. Acc. Chem. Res. 2019, acs.accounts.9b00064. https://doi.org/10.1021/acs.accounts.9b00064.

(19) Wang, H.; Xue, K.-F.; Yang, Y.; Hu, H.; Xu, J.-F.; Zhang, X. In Situ Hypoxia-Induced Supramolecular Perylene Diimide Radical Anions in Tumors for Photothermal Therapy with Improved Specificity. J. Am. Chem. Soc. 2022, 144 (5), 2360–2367. https://doi.org/10.1021/jacs.1c13067.

(25) Tang, F.; Wu, K.; Zhou, Z.; Wang, G.; Zhao, B.; Tan, S. Alkynyl-Functionalized Pyrene-Cored Perylene Diimide Electron Acceptors for Efficient Nonfullerene Organic Solar Cells. ACS Appl. Energy Mater. 2019, 2 (5), 3918–3926. https://doi.org/10.1021/acsaem.9b00611.

(29) Seo, J.; Khazi, M. I.; Kim, J.-M. Highly Responsive Triethylamine Vapor Sensor Based on a Perylene Diimide-Polydiacetylene System via Heat-Induced Tuning of the Molecular Packing Approach. Sensors and Actuators B: Chemical 2021, 334, 129660. https://doi.org/10.1016/j.snb.2021.129660.

Le mots « Devices » en position 86 est remplacé par le mot « agents »

(31) Gong, Q.; Xing, J.; Huang, Y.; Wu, A.; Yu, J.; Zhang, Q. Perylene Diimide Oligomer Nanoparticles with Ultrahigh Photothermal Conversion Efficiency for Cancer Theranostics. ACS Appl. Bio Mater. 2020, 3 (3), 1607–1615. https://doi.org/10.1021/acsabm.9b01187.

(33) Arantes, J. T.; Lima, M. P.; Fazzio, A.; Xiang, H.; Wei, S.-H.; Dalpian, G. M. Effects of Side-Chain and Electron Exchange Correlation on the Band Structure of Perylene Diimide Liquid Crystals: A Density Functional Study. J. Phys. Chem. B 2009, 113 (16), 5376–5380. https://doi.org/10.1021/jp8101018.

(34) Polkehn, M.; Tamura, H.; Eisenbrandt, P.; Haacke, S.; Méry, S.; Burghardt, I. Molecular Packing Determines Charge Separation in a Liquid Crystalline Bisthiophene–Perylene Diimide Donor–Acceptor Material. J. Phys. Chem. Lett. 2016, 7 (7), 1327–1334. https://doi.org/10.1021/acs.jpclett.6b00277.

Lines 90-91: « morphology of the nanostructures » was changed to « nano- and mesoscopic structures and liquid-crystalline compounds ».

In the conclusion and perspectives section, the author needs further condense and summarize.

Conclusion and perspectives:

As requested by the reviewer, we significantly condensed and summarized the conclusion and perspectives. 

Reviewer 2 Report

The review by Nadjib Kihal, Ali Nazemi, and Steve Bourgault is devoted to an interesting subject i.e. obtaining supramolecular nanostructures with tailored functional properties via a self-assembly process. Perylenediimide (PDI) is considered as a building lock block for the construction of supramolecular aggregates having big practical potential. The review is interesting and informative.

The authors did big efforts to provide a comprehensive overview of PDI-bioconjugates and used the most recent works on this subject. They focused on the synthesis and the relationships existing between the physicochemical properties of the bioconjugates, the conditions of self-assembly, and the resulting supramolecular structures. I have not any comments on the content and design of the review. I think it will be of interest to many readers. This allows me to recommend the manuscript for publication in its current form without additional changes.

Author Response

We thank reviewer 2 for his very positive comments.

Reviewer 3 Report

A class of perylenediimides is one of the most useful building blocks owing to their π–π stacking ability, photochemical/thermal stability and photophysical characteristics. Substituents with biological nature (e.g., peptides and oligonucleotides) have been exploited to decorate perylenediimines through their hydrogen-bonding interactions, chirality and so forth. This manuscript reviews recent advances in supramolecular assemblies based on perylenediimide bioconjugates. It is well organized and balanced. The reviewer suggests that it is publishable in the Journal. Minor self-revising is recommended, for example, chemical naming based on the current IUPAC rules, mixing of 1- and 3-letter symbols of amino acids and consistency the entire manuscript.

Author Response

We thank the reviewer 3 for his positive comments and fruitful suggestions. As requested by the reviewer 3, we included the following corrections:

Line 172 « Asp » was replaced by « D » and, « Tyr » by « Y ».

Lines 212-213: « Glu » was replaced by « glutamic acid»

Lines 232-233: « an asymmetric amphiphilic N-(tetra (L-alanine) glycine)-N′-(1-undecyldodecyl)-perylene-3,4,9,10-tetracarboxyl diimide » was replaced by « N-(tetra (L-alanine) glycine)-N′-(1-undecyldodecyl) functionalized perylene-3,4,9,10-tetracarboxyl diimide was designed as asymmetric amphiphilic derivative ».

Line 262: « Lys-Pro-Ala » was replaced by « KPA».

Line 277: «Glu» was replaced by « glutamic acid ».

Line 285: « FF » was replaced by « FF ».

Line 304: « Tyr, Trp and Phe » was replaced by « Y, W and F ».

Line 308: « Tyr » was replaced by « tyrosine ».

Lines 338-339: « N-Carbobenzyloxy-L-lysine-functionalized 1,6,7,12-tetrachloro-PDI » was replaced by « N,N’-α-Carbobenzyloxy-L-lysine-functionalized 1,6,7,12-tetrachloroperylene diimide ».

Line 372: « Ala, His, Phe and Val » was replaced by « A, H, F, and V ».

Line 384: « Phe » was replaced by « phenylalanine ».

Line 389: « His » was replaced by « Histidine ».

Line 410: « Thr and Asp » was replaced by « T and D ».

Line 412-413: « N-(tetra (L-alanine) glycine)-N′-(1-undecyldodecyl)-perylene-3,4,9,10-tetracarboxyl diimide» was replaced (corrected) by « N,N’-α-Carbobenzyloxy-L-lysine-functionalized 1,6,7,12-tetrachloroperylene diimide ».

Round 2

Reviewer 1 Report

Accept.